# Edible Bird’s Nest (EBN) Ameliorates the Effects of Indomethacin (IMC)-Induced Embryo Implantation Dysfunction in Rats

**DOI:** 10.3390/biology14020159

**Published:** 2025-02-04

**Authors:** Maria Amir, Nurhusien Yimer, Mark Hiew, Md Sabri Mohd Yusoff, Sadiq Mohammed Babatunde, Abdul Quddus

**Affiliations:** 1Department of Veterinary Clinical Studies, Faculty of Veterinary Medicine, Universiti Putra Malaysia, Serdang 43400, Selangor, Malaysia; gs50627@student.upm.edu.my (M.A.); mark@upm.edu.my (M.H.); 2Department of Physiology and Biochemistry, Faculty of Veterinary and Animal Sciences, Ziauddin University, Karachi 75600, Sindh, Pakistan; 3Department of Veterinary Sciences, School of Medicine, International Medical University, Bukit Jalil 57000, Kuala Lumpur, Malaysia; 4Department of Veterinary Reproduction, Faculty of Veterinary Medicine, Universitas Airlangga, Surabaya 60115, East Java, Indonesia; 5Department of Veterinary Pathology and Microbiology, Faculty of Veterinary Medicine, Universiti Putra Malaysia, Serdang 43400, Selangor, Malaysia; mdsabri@upm.edu.my; 6Department of Farm and Exotic Animal Medicine and Surgery, Faculty of Veterinary Medicine, Universiti Putra Malaysia, Serdang 43400, Selangor, Malaysia; babatunde@upm.edu.my; 7Faculty of Veterinary and Animal Science, Lasbela University of Agriculture Water and Marine Science, Uthal 90150, Balochistan, Pakistan; abdulquddus@luawms.edu.pk

**Keywords:** Edible Bird’s Nest, embryo implantation rate, fertility, indomethacin, uterine tissue, uterine toxicity

## Abstract

Indomethacin (IMC) is a nonsteroidal anti-inflammatory drug (NSAID) and a pharmaceutical product that is embryotoxic and leads to disruption of the implantation process when administered at doses higher than the therapeutic limit. IMC can cause infertility in animals and humans through inhibition of cyclo-oxygenase enzymes (COX), which are important in the synthesis of implantation mediating prostaglandins, and disruption of growth factors and steroid hormone receptors. Our results revealed that rats administered with a higher dose of EBN exhibited remarkable protective effects against IMC-induced toxicity on fertility and reproductive performance, thus highlighting the role of an edible bird’s nest in mitigating the impact of IMC on uterine tissues.

## 1. Introduction

One of the necessary conditions for mammalian reproduction is the efficient reciprocal contact between the implantation-competent blastocyst and the receptive uterus [1]. This is crucial for embryo implantation at the molecular and cellular levels. The synchronized development of preimplantation embryos to the blastocyst stage, the blastocyst’s escape from the zona pellucida, and the uterus differentiation into the receptive state of ovulation are all necessary stages for implantation [2].

The physiological process of reproduction can be interrupted by NSAIDs; for example, indomethacin (IMC) can cause infertility issues. IMC is known for its potent analgesic, anti-inflammatory, and fever-reducing effects. After oral administration, IMC is rapidly absorbed from the gastrointestinal tract, and its bioavailability is over 100%, with peak plasma concentrations occurring after a single dose. Adverse effects from therapeutic doses of IMC occur in 30–60% of people [3]. Women of reproductive age are more vulnerable to IMC exposure due to its high protein binding, limited gastrointestinal absorption and disruption of the blood–brain barrier [3,4].

Antioxidant imbalance and oxidative stress have been associated with the presence of reactive oxygen species, nitrogen oxides, and lipid peroxidation markers like malondialdehyde, all of which are exacerbated by lipid peroxidation inhibitors such as glutathione. As a result, research into the genetic and pharmacological processes of IMC-induced reproductive impairment is pertinent [5,6]. IMC may increase the risk of developing severe systemic toxicity without warning signs, primarily when used over a long period of time. IMC elicits lipid peroxidation and the formation of free radicals, with deleterious effects on tissues and causing oxidative tissue damage. Specifically, IMC was recently found to be toxic to the kidneys, liver, and intestine depending on the amount and duration of usage [5]. There is also a strong correlation between IMC exposure and infertility in dairy animals, as well as malnutrition and ovulatory or hormonal abnormalities [7].

For decades, people have ingested Edible Bird’s Nests (EBNs), which are created from the salivary secretions of swiftlet birds, as a healthful diet or tonic [8]. EBN has demonstrated significant medical effects in mice and rats by increasing the epidermal growth factors [8,9,10]. Researchers are becoming increasingly interested in the use of EBN, which has nutritional value (water-soluble protein, carbohydrate, iron, inorganic salt, and fiber) and numerous medicinal properties (anti-aging and antioxidant) for protection against heavy toxins [8,11,12].

A study on female rats found that EBN enhances fertility and embryo implantation by improving the differentiation and proliferation of uterine tissues, as evidenced by increased steroid receptor expressions [13]. Maternal hormones, especially estrogen and progesterone, and their receptors in the endometrium are required for successful implantation [14]. This leads to the production of signaling molecules like proliferating cell nuclear antigen (PCNA) and growth factors such as vascular endothelial growth factor (VEGF), epidermal growth factor (EGF), and their receptors [13,15]. EBN has been linked to a variety of therapeutic properties such as enhancing sexual ability or reproduction, but there is limited evidence-based data to support these suggested promising qualities. Thus, it remains unknown if EBN pretreatment is protective against IMC toxicity in the uterine tissue, with a focus on the uterine glandular epithelium (GE) and luminal epithelium (LE), expression of EGF, VEGF, and PCNA, and steroid receptors. This study investigated the effects of EBN pretreatment on IMC-induced toxicity in pregnant rats.

## 2. Materials and Methods

### 2.1. Animal and Experimental Design

A total of 30 adult (6 × 5 groups) female Sprague-Dawley rats (weight: 130–135 g) were placed in groups within the cages from Animal Resource Centre, A-Sapphire Enterprise. The acclimatization period lasted 8 days, where rats had access to water and a standard rat diet (A-Sapphire Enterprise). After acclimatization, the rats were randomly allocated into five groups and subjected to 8 weeks of in vivo EBN supplementation, followed by administering IMC via subcutaneous route (Table 1). The EBN was administered orally using gavage and the doses were determined based on the body weight of the rats [13]. The rats’ body weight gain (bwt) was measured weekly during the treatment. On 7 days post-implantation, the animals were sedated with ketamine and xylazine. A 23-gauge needle and a 3 mL syringe were utilized for clinical biochemistry and hematology.

### 2.2. Ethical Approval

All experiments followed the guidelines of the Universiti Putra Malaysia (UPM)-Institutional Animal Care and Use Committee (Project permission number: UPM/IAUC/AUR09/2016).

### 2.3. EBN Preparation

EBN was purchased from Mobile Harvesters (M) Sdn Bhd (House Bird’s Nest; batch no: 1802030650) and kept between 25 to 32 °C. The EBN sample conforms to the Malaysian Standards—Edible Bird’s Nest (EBN)—Specification (MS2334:2011). The EBN nests were prepared according to Chinese customs and collected from local EBN providers. In total, 1 g of EBN powder was dissolved in 100 mL of distilled water and heated to 56 °C for 45 min to make the EBN solution, as described by Albishtue et al. [13]. Specifically, the experimental design and preparation of EBN were adopted from the trial design used by Albishtue et al. [14] in the same species. The prepared EBN was then administered to rats at room temperature, with doses estimated based on the body weight.

### 2.4. Estrus Synchronization and Cycle Phases

PAP-stained daily vaginal smears were collected during the experiment at 09:00–10:00 h using cotton-tipped wooden swab sticks. After dipping the tips into the distilled water, they were carefully pushed into the vagina to a depth of approximately 1 cm. Then, using a rotating motion, the swab’s tip was used to smear the sample onto a clean microscope slide (75 mm × 25 mm). The percentage of cells in the vaginal smear was determined by fixing the slide in methanol, which was stained with 5% Giemsa stain for 3 min and viewed under a light microscope (Leitz Laborlux-S, Wetzlar, Germany), where all cell counts were made at 200× magnification. Micrographs were taken and each cell type (i.e., nucleated, leukocytes, and cornified cells) was counted. After counting cells for designated animals, the percentage of each cell type was determined for all of the animals during each stage of the estrous cycle [16]. The length of the estrous cycle in mice is between 4–5 days (proestrus: <24 h, estrus: 12–48 h, metestrus; 8–24 h, and diestrus: 48–72 h) [17,18]. During estrus synchronization, two doses of 0.5 μg of cloprostenol were injected intraperitoneally into the rats three days before the experiment, at which point the rats’ estrus cycles were synchronized.

### 2.5. Macroscopic and Microscopic Inspections of the Uterus

One female rat and one male rat were placed in the reproductive cage in the evening at 8 h to test the fertility of both genders. Then, in the morning, the female rats were tested to observe a vaginal plug to confirm the male and female rats’ mating, after which vaginal smears were performed and stained with blue methylene to observe sperm in them. The day on which sperm or a vaginal plug were observed in the female rats was considered day zero of pregnancy [19]. On day 7 of pregnancy, the uteri were removed and weighed after sacrificing the rats. Meanwhile, the implantation rate and fertility index were studied in the uterus. The fertility index was calculated using the formula: number of pregnant females/number of animals copulated *100) [13].

The Medical Image Analysis microscope was used to discover any histological alterations (Motic Image plus 2.0, PhysLab, (Xiamen, China). Tissue samples were cut 4 μm and fixed in 10% formalin for 24 h, segmented, and stained with hematoxylin and eosin (HE). Specifically, the weight of the uterine tissues was measured using the following equation cited by [13]:Uterine weight (g)/body weight (g) ratio = weight of uterus divided by bwt of animal (g) × 100

By measuring the distance along a transverse section of the proximal and distal ends of the endometrium, which is visually regarded as the thickest of any single site, the endometrial thickness (from the basement membrane to the apical surface of the epithelium) can be ascertained. Each sample was measured, with five fields of view obtained for each region, and the average value was considered [20].

### 2.6. Scoring Method of H&E

Three slides from each organ were observed for scoring. Five microscopic areas with different magnifications were observed for each slide (200× and 400×). The numbers and percentage of inflammatory cells, degeneration of luminal epithelium (LE) and glandular epithelium (GE), congestion, and vacuolation were quantified in all the groups using an image analyzer software (Motic Images Software Plus 2.0, PhysLab, (Xiamen, China). The changes in the treated and control groups were scored on a scale of 0–3. Score 0 = no lesions, 1 = less than 30% of the lesions seen, 2 = less than 60% of the lesions seen, and 3 = more than 60% of the lesions seen. The pathologist assessing the sections was blinded to the treatment received by the rats [13].

### 2.7. Sample Preparation for SEM

The samples were rinsed thrice for 10 min in 0.1 M sodium cacodylate buffer after 16–24 h in 4% glutaraldehyde at 4 °C. Subsequently, the samples were dehydrated in a graded series of acetone 35%, 50%, 75%, and 90%, for 10 min each, and 100% for three changes of 15 min each. The samples were then dried in a critical point dryer (Baltic CPD, Wetzlar, Germany) for 30 min using liquid CO_2_ as the traditional fluid. Next, the tissue was mounted on aluminum stubs using double-sided tape and sputter coated with a 12 nm golden palladium in a sputter coater (SCD005, (Wetzlar, Germany). All the prepared specimens were viewed under a Scanning Electron Microscope (SEM) (Philips XL 30ESEM, The Netherlands) operating at 15 keV [21]. FIJI ImageJ2 software 2.9.0 was used to analyze the expression of the pinopodes in the uterine endometrium [22]. Using FIJI software, the diameter of the pinopodes was measured [23].

### 2.8. Expressions of EGF, EGFR, PCNA, VEGF, E2R, and P4R in Uterine Tissues by Immunohistochemistry

Protein expression was examined by immunohistochemistry. The specimens were sliced to a thickness of 4 μm to dehydrate and deparaffinized the uterine tissue. The antigens were recovered by incubating the samples at 80 °C for 10 min in 0.01 M citrate buffer (pH 6.0), followed by cooling at room temperature. Thereafter, the samples were identified using a Dako pen after rinsing for 1 min in Tris-buffered saline (TBS) (Dako, Glostrup, Denmark) and the surplus wash buffer was tapped out. The specimen was treated with a peroxidase block solution (0.03% hydrogen peroxide containing sodium azide) from the Animal Research Kit (Code K3954, Dako, CA, USA) and incubated for 10 min at 37 °C; it was then rinsed with Tris-buffered saline (TBS).

The samples were then incubated with rabbit anti-E2R (Anti-Estrogen Receptor alpha antibody [E115]-cHip, grade ab32063, 1:200), rabbit anti-P4R (Anti-Progesterone Receptor antibody [SP2], ab16661, 1:100), rabbit anti-EGF (Anti-EGF antibody ab77851, 1:100), and rabbit anti-EGFR (Anti-EGFR antibody [EP38Y], ab52894, 1:500), all manufactured by Abcam, USA, in TBS for 40 min at 37 °C. As a negative control, a section was also prepared by using TBS instead of the main antibodies. After being washed in TBS, the samples were incubated at 37 °C for 40 min with biotin-labeled secondary antibody (Envision System-HRP Labeled Polymer Anti-Rabbit (Code K4002), Dako, CA, USA) or mouse anti-PCNA antibody PC-10 (Code M0879, Dako, Glostrup, Denmark) or mouse VEGF antibody (Anti-VEGF antibody [v-1] ab1316, Abcam, MA, USA). Streptavidin-peroxidase (Animal Research Kit, Code K3954, Dako, CA, USA) was applied to the tissue sections for 20 min at 37 °C. DAB (3,30-diaminobenzidine) stain was applied to perform chromogenic staining for 5 min after washing the sections in distilled water. Afterward, a counterstain was used to highlight the stained regions (hematoxylin). A good reaction was defined as a yellowish-brown staining color detected under a microscope (magnification 400×), with the density of the staining. The percentage of positive epithelial cells was graded as follows: no stained cells (Grade 0), 1–25% stained cells (Grade 1), 26–50% stained cells (Grade 2), 51–75% stained cells (Grade 3), and 76–100% stained cells (Grade 4) in the representative field. The approximate percentage of immunopositively cells was analyzed based on 10 representative fields. The immunostaining intensity was scored as negative (0), weak (1), moderate (2), or strong/intense (3) [24]. The total index was calculated using the following formula: total index = [percentage of positive cells] × [immunostaining intensity].

### 2.9. Statistical Analysis

All data were assessed for normality. Since the data confirmed the assumptions of normality using a Shapiro–Wilk test, a one-way analysis of variance ANOVA with a Bonferroni multiple comparison post-hoc test was applied to assess the difference in mean values of uterine to body weight ratios, percentage of fertility index, embryonic implantation rate, expression levels of growth factors, EGFR and PCNA, and steroid receptors, except for the body weight, in which a two-way ANOVA was used with a Bonferroni post-hoc test using GraphPad prism software 8.0.2. SEM images were analyzed using FIJI ImageJ2 software 2.9.0 and the data were analyzed by a one-way ANOVA followed by a post-hoc Bonferroni test using GraphPad prism software. Data were considered significantly different at *p* ≤ 0.05.

## 3. Results

### 3.1. Effect of EBN on Body Weight Gain (BWG)

During the 8 weeks of treatment with EBN, all rats showed normal (4–5 days) estrus cycles and no changes in the cell shapes of the vaginal smear. At week 8, the bwt was significantly decreased (264.6 ± 2.5 g, *p* ≤ 0.05) in the positive control (G2) in comparison with group G1 (283 ± 2.3 g) and EBN supplemented groups (G3: 278 ± 1.7 g; G4: 280 ± 1.8 g; G5: 296.5 ± 2.1 g), as shown in Figure 1. Significant differences were recorded in the rats’ bwt in all groups throughout the experimental period.

### 3.2. Effect of EBN on Uterine Body Weight Ratio (UBWR) and Thickness of Luminal Epithelium (LE)

The uterine bodyweight ratio (UBWR) was lowest (0.8 ± 0.05 g, *p* ≤ 0.05) in the positive control (G2) group and highest (*p* ≤ 0.05) in the negative control group and G5 treatments. The changes increased significantly in the treatment groups and attained the highest value (0.9 ± 0.02 g, *p* ≤ 0.05) in G5 group (Figure 2a). No significant differences were observed between G3 and G4. The result revealed that EBN enhanced uterine weight and length. The IMC-treated group (G2) (22.3 ± 3.9 g, *p* ≤ 0.05) had thinner luminal epithelium than the G5 (40 ± 2 g, *p* ≤ 0.05) group. In addition, EBN enhanced LE thickness dose-dependently. The treated groups G4 and G5 had considerably thicker LE compared to the untreated groups (Figure 2b).

### 3.3. Fertility Index and Embryo Implantation Rate

On day 8 of the pregnancy, the fertility index and the number of EIRs in the EBN-supplemented groups of rats were assessed and compared with control. Displayed in Figure 3 are the total number of embryo implantations, which looked like a string of small beads. Among all groups, G1 and G5 reflected the highest (10.6 ± 1.9 g and 14.6 ± 3.2 g, respectively, *p* ≤ 0.05) number of EIRs relative to G2, G3, and G4 (Figure 4a). Moreover, fertility index was observed to be highest in G5 compared to controls and other EBN treatment groups (Figure 4b).

### 3.4. Histopathological Examination of Uterine Tissue

No apparent gross pathological lesions were found in the uterus (Figure 5). In IMC-induced inflammatory alterations, atrophy of the endometrium, vacuolar degeneration, inflammatory alterations, and muscle atrophy were detected in endometrial epithelial cells. G4 and G5 exhibited the typical uterine structure as the control. The microscopic examination of the uteri also depicted remarkable development and greater endometrial structures in G5 samples (Table 2).

### 3.5. Uterine Examinations Using SEM

On Pd8, scanning electron microscopic investigations of control and EBN-supplemented rats indicated differences, such as smooth, few-folded pinopodes in uterine epithelium [25]. Meanwhile, the positive control (G2) demonstrated several folds and uneven pinopodes. However, the Control and G5 groups showed microvilli and a greater number of pinopodes compared to the G3 and G4 groups with a lower number of pinopodes (Figure 6a). While SEM image analysis allowed calculating the diameter of pinopodes. The analysis demonstrated significantly lower numbers in G2 (1.7 ± 0.2 g, *p* ≤ 0.05) and higher numbers in the G1 and G5 groups (2.7 ± 0.1 and 3.06 ± 0.1 g, respectively, *p* ≤ 0.05) compared to the rest of EBN-treated groups, as shown in (Figure 6b).

### 3.6. Expressions and Scoring of EGF, EGFR, PCNA, and VEGF in the Uterine Tissues

The G4 and G5 with the highest EBN dose reflected significantly higher expression of EGF, EGFR, PCNA, and VEGF in the uterine section compared with the G1 and control groups (*p* ≤ 0.05) (Table 3). The expression levels across the stroma cells (S), the glandular epithelium (GE), and the luminal epithelium (LE) were significantly higher than those in the control and G2 (Figure 7, Figure 8, Figure 9 and Figure 10). Furthermore, the increase in the dose of EBN treatment resulted in the gravid uterine, an increase in the density of stromal (S cells, LE and GE), and increased expression levels of EGFR, EGF, VEGF, and PCNA. Variations in the expression level and pattern were observed between cellular compartments and uterine regions.

For EGF, the expression started in the control group with a weak reaction involving a few S, GE, and LE cells that primarily revealed partial staining (the nucleus and/or cell membrane part). Nevertheless, the EGF expression level changed from low (as it did for G1 and G3 with low-dose EBN) to a strong staining reaction across all parts of the sections, as observed in G4 and G5 (Figure 7). In both G4 and G5, a relatively dense and strong immune reaction was observed at the GE. Meanwhile, EGFR and VEGF displayed a different scenario as no specific immune reactions were found in the G1, G2, or G3 slides (Figure 8 and Figure 9). VEGF and PCNA (Figure 10) reactions started to be seen at 90 mg/kg bwt of EBN supplement (G4) with a similar degree of expression and staining compared to G5, which was supplemented with the highest dose of EBN (*p* ≥ 0.05). The observed VEGF expression in G4 and G5 involved all areas of the S, GE, and LE (Figure 8) and some feeble reactions along the S and LE, respectively.

### 3.7. Expression of Steroid Hormone Receptors (E2R and P4R) in Uterine Tissues

The stromal cell density and the E2R expression (Figure 11) in S, GE, and LE increased with the EBN dose. G4 and G5 recorded a higher P4R expression (*p* ≤ 0.05) than G3, which demonstrated mild expressions while the lowest expressions were observed in G1 and G2 (Figure 12). The rats treated with the highest EBN doses (G5) depicted a significant increase in E2R expression compared with the control and the other treated groups (*p* = 0.01) (Table 4).

## 4. Discussion

The use of EBN as a protective agent against heavy toxins has recently received remarkable interest among researchers. The present study assessed the protective effects of EBN against IMC-induced toxicity in pregnant rats. IMC is a potent NSAID that was discovered during a search for anti-inflammatory and analgesic medicines [26]. Although the actual mechanisms for the anti-inflammatory and analgesic properties of IMC are not fully understood, previous studies suggest that IMC may have unique direct neuronal inhibitory mechanisms or nitric oxide-dependent inhibitory pathways in addition to the cyclooxygenase inhibitors they operate [11]. IMC has intermediate-to-long half-lives (4.5+ hours) [27] and several events may induce the release of IMC into the general circulation [28].

We found that IMC affected the uterine body weight ratio in the positive control (G2) group. This was further evident in the histological findings as the IMC-treated group had thinner luminal epithelium compared to the G5 group. As gleaned from this study, previous investigations have equally demonstrated deleterious effects of IMC in the female and male rat’s reproductive organs [29,30,31].

Alterations in the uterine body weight ratio might also indicate IMC’s effect on the endometrial glands. Histological examination of the positive control revealed damaged and endometrial glands evident by uterine lumen thickening, endometrial atrophy, vacuolar degeneration, inflammatory abnormalities, and muscle atrophy, which are consistent with histological changes reported in a previous study [32]. These histopathological changes may stem from the restriction of PGs imposed by the IMC, leading to vasoconstriction of the uterine blood vessels [33,34]. Subsequently, the loss of blood supply to the muscles may elicit smooth muscular atrophy [35]. Other studies also found that higher doses of IMC therapy reduced uterine horn, ancillary function, ciliary activity, and horn lumen [31,36]. IMC may damage uterine glands, resulting in altered endometrial glandular secretions such as enzymes, growth factors, cytokines, hormones, and transport proteins needed for conceptus formation, which may harm in utero physiological activities [37].

In this study, supplementation with higher doses of EBN (G4 and G5) ameliorated the IMC-induced toxicity in the uterus. Notably, G5 recorded the highest uterine body weight ratio, reflecting that EBN improved uterine weight and length. Morphological features of the endometrial gland and epithelia are important in determining the stages of the endometrial cycle [38]. The lumina epithelium is a narrow to medium-sized tissue, which increases during the secretory phase. Since the EBN-supplemented group exhibited thicker lumina epithelium coupled with increased height and length of the glands, it is an indication of secretory activities within the endometrium [39].

Overall, these results corroborate previous findings on the use of EBN in annulling the adverse effects of NSAIDs on rats’ uterus [5,21]. For instance, necrosis of uterine glands and associated lining cells caused by IMC toxicity was significantly reduced by EBN supplementation, evident by the significantly higher number of uterine glands and increased concentrations of SOD compared to the control groups [13]. In our previous study, restoration of uterine histomorphology was observed in rats following EBN supplementation [21]. Thus, the anti-inflammatory properties of EBN might have played an important role in protecting the uterine glands and lining cells against IMC-induced toxicity in this study. Moreover, experimentally induced inflammation and oxidative stress in rats were significantly attenuated by EBN treatment upon analyzing the total antioxidant status and inflammatory markers [40,41]. Although we did not investigate the concentration of NOS1 in the uterine muscles, the pretreatment with EBN might have inhibited NO production from macrophages, which is one of the main indicators of anti-inflammatory activity. EBN’s antioxidant properties may ameliorate IMC-induced toxicity by protecting the oxidatively sensitive uterine glands and lining cells and preventing the formation of reactive oxygen species [42,43].

On day 8 of pregnancy, the fertility index and EIR in the EBN-supplemented groups were significantly higher in relation to G2, G3, and G4. Additionally, the embryo implantation rate in EBN-supplemented rats was smooth with few-folded uterine epithelium compared to the positive control, which demonstrated several folds and uneven pinopodes. Similar findings were highlighted in rats supplemented with EBN, which recorded an increase in fertility and embryo implantation rate by stimulating the proliferation and differentiation of uterine structures [21].

Humans and rodents have different pinopodes in terms of size, shape, and substance. Pinopodes have a diameter of 3.0 to 4.0 μm in rats and 6 μm in mice and humans [44]. The current study showed a significantly higher average diameter of pinopodes in the endometrium in the groups of rats exposed to IMC and treated with the highest dose of EBN (120 mg/kg bwt) compared to the rest of other treated groups. This might partly explain the reason why G5 rats also showed a significantly higher embryo implantation rate compared to the rest of the experimental groups including the control, which in turn might imply the intense potential role of EBN as a supplement beyond mitigating the toxic effect of IMC. EBN also improved the growth of ultrastructural pinopods, thereby supporting the uterine epithelium during embryo attachment.

To elucidate the promising findings discussed above, the expressions of EGF, EGFR, PCNA, and VEGF in the uterine section of all the experimental groups were investigated. Results revealed that the groups with the highest EBN dose recorded significantly higher expressions of EGF, EGFR, PCNA, and VEGF in relation to G1 and the control groups. EGF is a strong mitogen, which is well-documented to enhance endometrial growth in humans, sheep, and rodents [45]. EGF and interleukin 6 are intercellular mediators that regulate cellular growth, survival, differentiation, and function [46]. Specifically, EGF expression facilitates uterine, ovarian, and uterine gonad proliferation, as well as endothelial thickness.

Meanwhile, PCNA acts as a processivity factor for DNA polymerase by encircling the DNA, thus serving as a scaffold to recruit proteins during the replication and repair of DNA and chromatin remodeling [47]. It also participates in post-replication repair, particularly in the DNA damage tolerance pathway [48]. VEGF is a vascular permeability factor, and an endothelial cell-specific mitogen synthesized by several cell types such as macrophages, keratinocytes, tumor cells, and renal mesangial cells. Apart from the vascular system, VEGF is required for the endometrium’s growth and maintenance, and it participates in normal physiological functions like wound healing, hematopoiesis, and development [49,50]. VEGF increases blood flow to the uterine glands and stromal cells, indicating that high VEGF levels were promoted by EBN supplementation, leading to the proliferation of lumina and glandular epithelium [50]. Therefore, the increased expression of these factors reflects the possible pathways through which EBN mitigated IMC-induced toxicity in the pregnant rats’ uterine glands and lining cells. Increased EGF, VEGF, and PCNA expressions imply that EBN has proliferative and therapeutic effects against IMC toxicity [37]. EBN was also shown to demonstrate an EGF-like activity and enhanced expressions of EGF and its receptor in rats’ endometrium [21,41]. These findings suggest that EBN affects reproduction and fertility.

Another important finding is the variation in the expression level and pattern between the cellular compartments and uterine regions. For instance, while VEGF expression was prominent in all areas of the stroma cells, glandular epithelium, and lumina epithelium in groups with the highest dose of EBN, only mild expressions were observed in the G2 and G3, respectively. This result further highlights the toxicity of IMC on rats’ uterine tissues as well as the dose-dependent effect of EBN in protecting the glands from further damage.

We also found that the expressions of uterine estrogen and progesterone receptors (E2R and P4R) were very mild in the positive control group, but both receptors increased significantly with higher doses of EBN. Moreover, ovarian estrogen and progesterone, and their receptors in the endometrium, are required for successful implantation [47]. Thus, the increased expression of E2R and P4R in groups supplemented with higher doses of EBN might have contributed to the higher fertility index and a greater number of embryo implantation rates observed earlier. Apart from protecting against IMC-induced toxicity, these events will enhance the fertility and reproductive performance of the studied rats.

The protective roles of EBN can also be gleaned from previous related studies. EBN was demonstrated to be neuroprotective against estrogen-deficiency-related aging [51]. EBN blocks caspase-3 activation to reduce embryonic neuronal death [52]. Previous research on EBN’s composition identified testosterone, E2, P4, LH, FSH, and prolactin as important constituents [46]. Notably, a study found that the synergic actions between VEGF, FSH, and estrogen led to the prevention of programmed cell death by blocking caspase 3 and stimulating cell growth [53,54]. EBN is, thus, enriched with several beneficial constituents similar to EGF that facilitate cell division, proliferation, and tissue regeneration, reflecting its diverse protective effects [53].

## 5. Conclusions

In conclusion, the supplementation of EBN in female rats could ameliorate IMC-induced reproductive toxicity and have promising activity toward reproductive performance. IMC groups exhibited a significant decrease in body weight compared to the treatment groups. Groups treated with IMC and EBN at 120 mg/kg dosage had the highest EIR. A high dose of EBN elicited a protective effect on uterine tissue over IMC-induced inflammatory alterations, atrophy of the endometrium, and vacuolar degeneration. EBN demonstrated tissue proliferating effects by enhanced expressions of growth factors and receptors of steroid hormones such as EGF, VEGF, EGFR, PCNA, E2R, and P4R in uterine tissues, respectively. Moreover, supplementation of EBN augments the functions of the reproductive tissues and protects against reproductive toxicity associated with IMC, which was evident by improved fertility indexes recorded in supplemented rats. As such, EBN supplements might also assist in preserving the fertility of individuals who depend on IMC for the treatment of chronic inflammatory illness, though future studies would be required to verify this.

## Figures and Tables

**Figure 1 biology-14-00159-f001:**
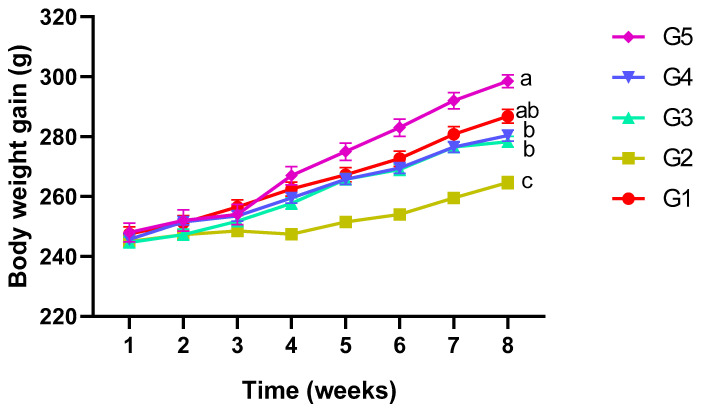
Effects of IMC and EBN administration on body weight after pregnancy. Data are expressed as Mean ± SEM. Error bars with different letters, i.e., a, b, and c, along the lines denote significant differences at *p* ≤ 0.05 (n = 6 rats per group). G1 = Control, G2 = (IMC 4.33 mg/kg bwt), G3 = (IMC 4.33 + EBN 60 mg/kg bwt), G4 = (IMC 4.33 + EBN 90 mg/kg bwt), G5 = (4.33 + 120 mg/kg bwt).

**Figure 2 biology-14-00159-f002:**
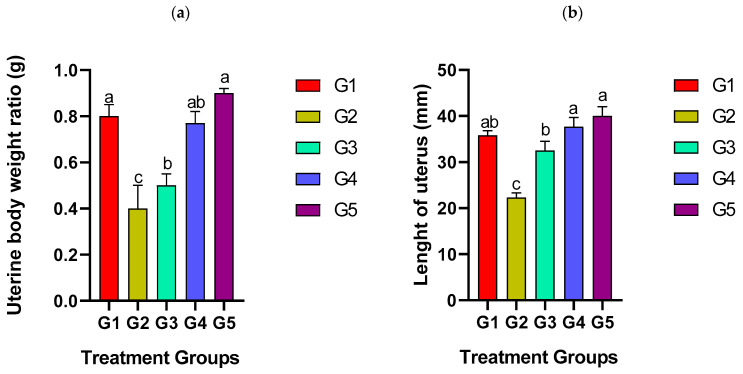
Effects of IMC and EBN administration on (**a**) uterine body weight after pregnancy; (**b**) thickness of LE. Data are expressed as Mean ± SEM. Error bars with different letters, i.e., a, b, and c, along the rows denote significant differences at *p ≤* 0.05 (n = 6 rats per group). G1 = Control, G2 = (IMC 4.33 mg/kg bwt), G3 = (IMC 4.33 + EBN60 mg/kg bwt), G4 = (IMC 4.33 + EBN 90 mg/kg bwt), G5 = (4.33 + 120 mg/kg bwt).

**Figure 3 biology-14-00159-f003:**
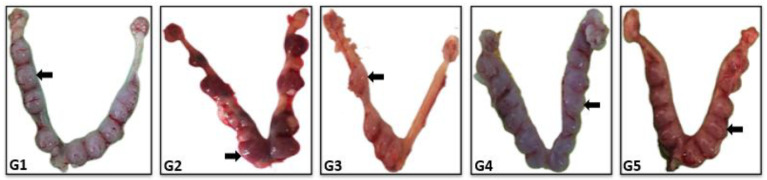
A representative photograph of implanted blastocysts in uteri (black arrow). EBN supplementation resulted in an increased number of embryo implantations (n = 6 rats per group). G1 = Control, G2 = (IMC 4.33 mg/kg bwt), G3 = (IMC 4.33 + EBN 60 mg/kg bwt), G4 = (IMC 4.33 + EBN 90 mg/kg bwt), G5 = (4.33 + 120 mg/kg bwt).

**Figure 4 biology-14-00159-f004:**
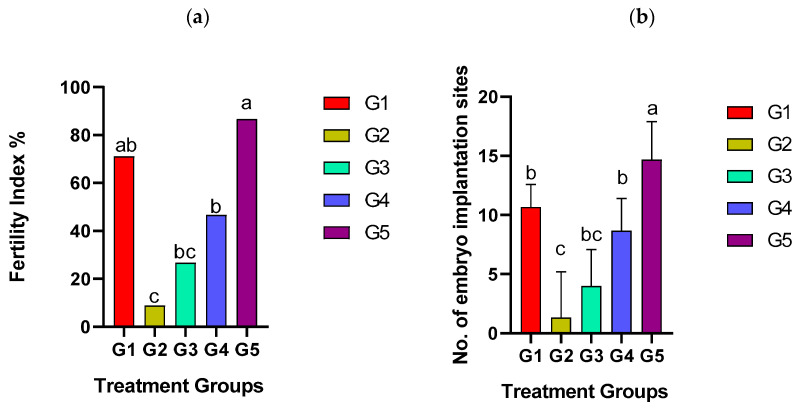
Effects of IMC and EBN on fertility index % (**a**) and embryo implantation rate (**b**). Data are expressed as Mean ± SEM. Error bars with different letters, i.e., a, b, and c, along the rows denote significant differences at *p ≤* 0.05 (n = 6 rats per group). G1 = Control, G2 = (IMC 4.33 mg/kg bwt), G3 = (IMC 4.33 + EBN 60 mg/kg bwt), G4 = (IMC 4.33 + EBN 90 mg/kg bwt), G5 = (4.33 + 120 mg/kg bwt).

**Figure 5 biology-14-00159-f005:**
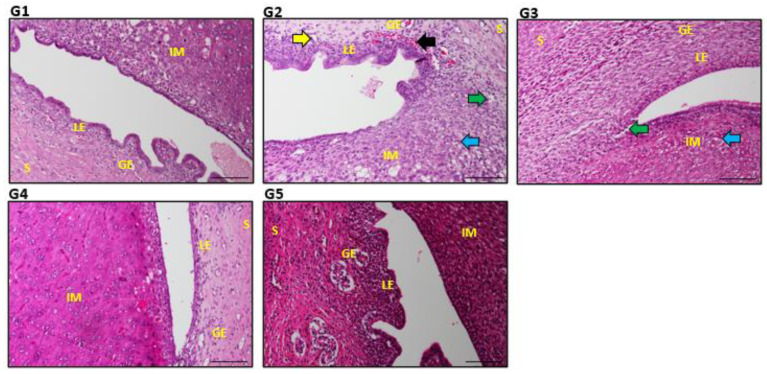
Impact of IMC exposure and EBN therapy on uterine histomorphology. LE signifies the luminal epithelium, S the stroma, GE the glandular epithelium, and IM implantation sites. Normal uterine glands are indicated by yellow arrows, but congestion is indicated by a black arrow, vacuolation is indicated by a blue arrow, and endometrial atrophy by a green arrow in G2 and G3. Deterioration of LE cells by a black arrow is manifested in G2 compared to treatment groups. H&E stains. Scale bar 100 μm (Appendix A).

**Figure 6 biology-14-00159-f006:**
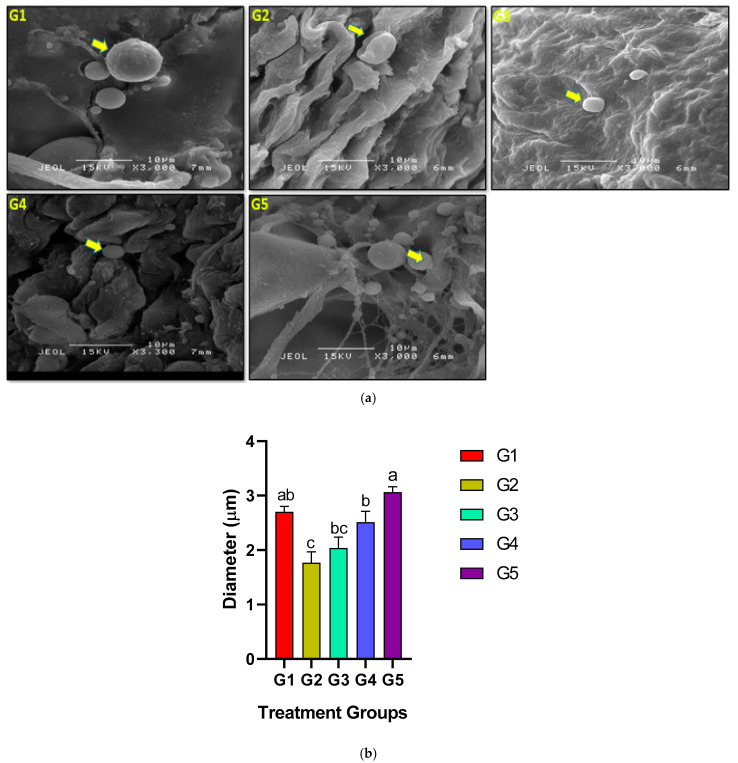
(**a**) SEM micrographs of uteri for pregnant rats. The yellow arrow indicates pinopodes and the green arrow indicates microvilli (scale bar 10 μm) with smooth and minor folded uterine epithelium in G5 compared to the multiple folds and uneven pinopodes observed in G1, G2, and G3. (**b**) Effects of IMC and EBN administration on the quantitative analysis of pinopodes diameter in the endometrium. Data are expressed as Mean ± SEM. Error bars with different letters, i.e., a, b, and c, along the rows denote significant differences at *p* ≤ 0.05 (n = 6 rats per group). G1 = Control, G2 = (IMC 4.33 mg/kg bwt), G3 = (IMC 4.33 + EBN 60 mg/kg bwt), G4 = (IMC 4.33 + EBN 90 mg/kg bwt), G5 = (4.33 + 120 mg/kg bwt).

**Figure 7 biology-14-00159-f007:**
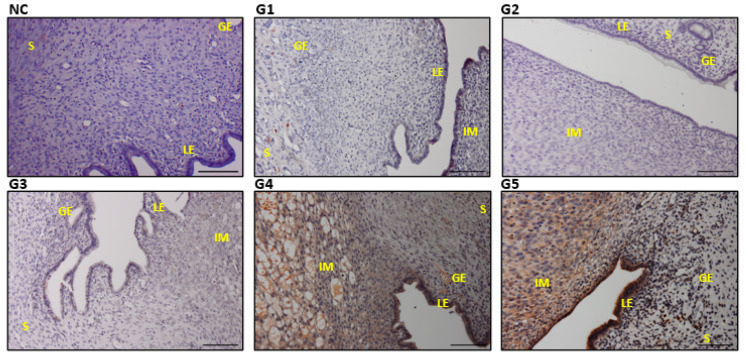
Photomicrograph sections of the uteri of rats of different experimental groups (G1, G2, G3, G4, and G5) treated with different doses of EBN showing expressions of epidermal growth factor (EGF). stroma (S), glandular epithelium (GE), and luminal epithelium (LE), with implantation sites (IM). Scale bar 100 μm (Appendix A).

**Figure 8 biology-14-00159-f008:**
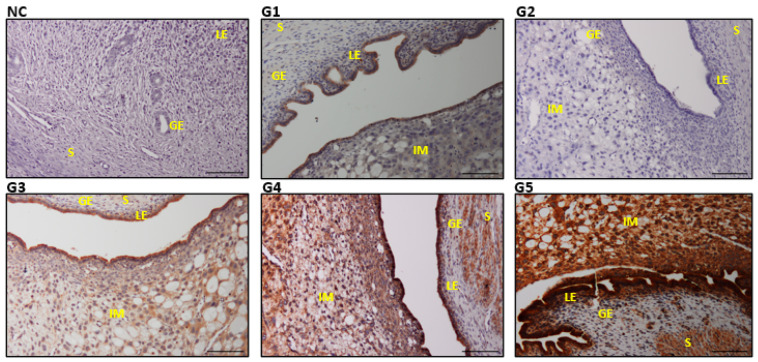
Photomicrograph sections of the uteri of rats of different experimental groups (G1, G2, G3, G4, and G5) treated with different doses of EBN showing expressions of EGFR. Stroma (S), glandular epithelium (GE), and luminal epithelium (LE), with implantation sites (IM). Scale bar 100 μm (Appendix A).

**Figure 9 biology-14-00159-f009:**
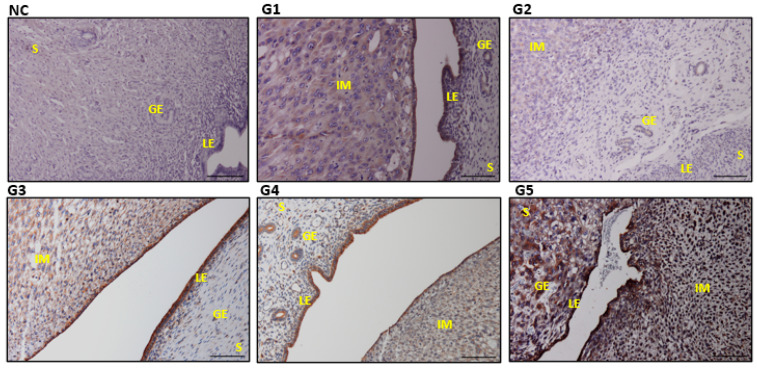
Photomicrograph sections of the uteri of rats of different experimental groups (G1, G2, G3, G4, and G5) treated with different doses of EBN showing expressions of vascular endothelial growth factor (VEGF). Stroma (S), glandular epithelium (GE), and luminal epithelium (LE), with implantation sites (IM). Scale bar 100 μm (Appendix A).

**Figure 10 biology-14-00159-f010:**
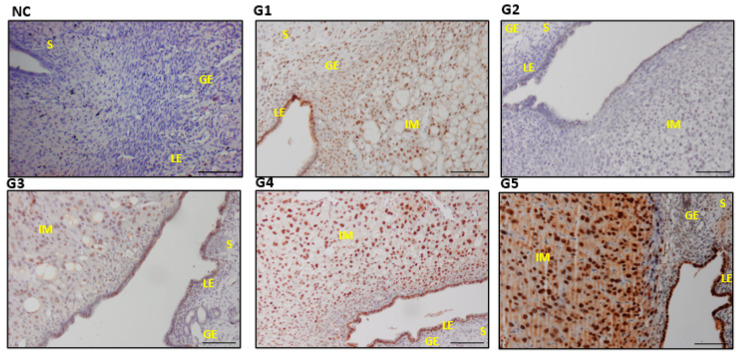
Photomicrograph sections of the uteri of rats of different experimental groups (G1, G2, G3, G4, and G5) treated with different doses of EBN showing expressions of proliferating cell nuclear antigen (PCNA). Stroma (S), glandular epithelium (GE), and luminal epithelium (LE), with implantation sites (IM). Scale bar 100 μm (Appendix A).

**Figure 11 biology-14-00159-f011:**
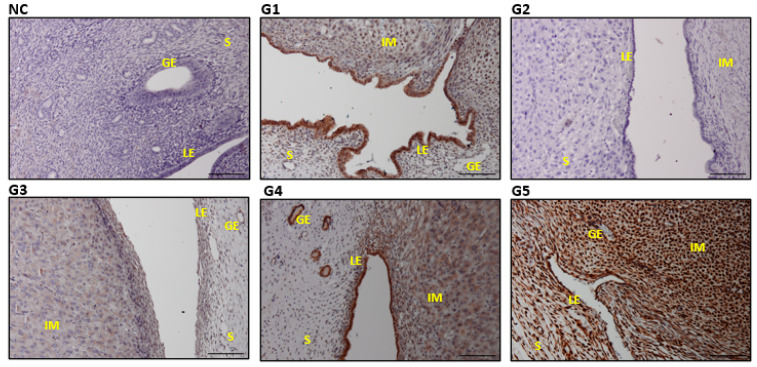
Photomicrograph sections of rat uteri of different experimental groups (G1, G2, G3, and G4). Estrogen receptor (E2R) was observed in all groups with the highest expression in G4 and G5. Stroma (S), glandular epithelium (GE), and luminal epithelium (LE), with implantation sites (IM). Scale bar 100 μm (Appendix A).

**Figure 12 biology-14-00159-f012:**
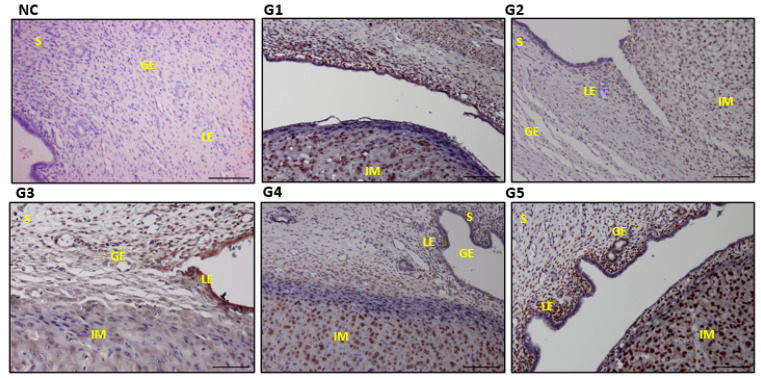
Photomicrograph sections of rat uteri of different experimental groups (G1, G2, G3, and G4) showing progesterone receptor (P4R) expressions. Note the higher expression of P4R in G5. Stroma (S), glandular epithelium (GE), and luminal epithelium (LE), with implantation sites (IM). Scale bar 100 μm (Appendix A).

**Table 1 biology-14-00159-t001:** Group allotment of animals and treatment administration.

Group	Assignment Rats/Group)	(Type of Treatment (Dose)/8 Weeks
Control	G1 (Control)	Normal diet + Normal saline
Treated Groups	G2 (Positive control)	IMC only (4.33 mg/kg)
	G3 (Low dose)	IMC + EBN (4.33 mg/kg + 60 mg/kg bwt.)
	G4 (Medium dose)	IMC + EBN (4.33 mg/kg + 90 mg/kg bwt.)
	G5 (High dose)	IMC + EBN (4.33 mg/kg +120 mg/kg bwt.)

IMC = Indomethacin, EBN = Edible Bird’s Nest, bwt = body weight.

**Table 2 biology-14-00159-t002:** Differences in mean score of histopathological lesions in the uteri of rats exposed to IMC with EBN supplements.

Parameters	Groups				
	G1	G2	G3	G4	G5
Vacuolation	0 ± 0 ^a^	2.8 ± 0.1 ^c^	1.6 ± 0.3 ^b^	0.5 ± 0.2 ^a^	0 ± 0 ^a^
Atrophy	0 ± 0 ^a^	2.3 ± 0.3 ^c^	2 ± 0 ^c^	0.6 ± 0.3 ^a^	0 ± 0 ^a^
Infiltration of inflammatory cells	0 ± 0 ^a^	2.5 ± 0.2 ^c^	2 ± 0 ^c^	0.8 ± 0.1 ^a^	0.5 ± 0 ^a^
Congestion	0 ± 0 ^a^	3 ± 0 ^c^	2.3 ± 0.3 ^c^	0 ± 0 ^a^	0 ± 0 ^a^

Note: Means with different superscripts a, b, and c within the same row are significantly different from each other at *p ≤* 0.05. Data are expressed as Mean ± SEM (n = 6 rats per group). G1 = Control, G2 = (IMC 4.33 mg/kg bwt), G3 = (IMC 4.33 + EBN 60 mg/kg bwt), G4 = (IMC 4.33 + EBN 90 mg/kg bwt), G5 = (4.33 + 120 mg/kg bwt).

**Table 3 biology-14-00159-t003:** Mean scores of EGF, EGFR, VEGF, and PCNA expression in the uterine tissues of the rats exposed to IMC toxicity and augmented with EBN.

Parameters	Groups				
	G1	G2	G3	G4	G5
EGF in LE	1 ± 0 ^c^	0.5 ± 0 ^c^	0.5 ± 0 ^c^	1.5 ± 0 ^b^	3 ± 0 ^a^
EGF in GE	1 ± 0 ^c^	0.5 ± 0 ^c^	0.5 ± 0 ^c^	1.5 ± 0 ^b^	3 ± 0 ^a^
EGF in S	0 ± 0 ^c^	0 ± 0 ^c^	0 ± 0 ^c^	1 ± 0 ^c^	2.5 ± 0 ^a^
VEGF in LE	2 ± 0 ^b^	0 ± 0 ^c^	1.16 ± 0.16 ^b^	2.5 ± 0 ^b^	2.5 ± 0 ^a^
VEGF in GE	1 ± 0 ^c^	0 ± 0 ^c^	2 ± 0 ^b^	2.5 ± 0 ^b^	2.5 ± 0 ^a^
VEGF in S	1.3 ± 1.6 ^b^	1 ± 0 ^c^	2.5 ± 0 ^b^	2.5 ± 0 ^b^	2.5 ± 0 ^a^
EGFR in LE	1.5 ± 0 ^b^	1 ± 0 ^c^	1 ± 0 ^c^	2.5 ± 0 ^b^	3 ± 0 ^a^
EGFR in GE	1 ± 0 ^c^	0.6 ± 0.3 ^c^	1 ± 0 ^c^	1.5 ± 0 ^b^	3 ± 0 ^a^
EGFR in S	1.3 ± 0.3 ^b^	0.3 ± 0.16 ^c^	1.5 ± 0 ^b^	1.5 ± 0 ^b^	3 ± 0 ^a^
PCNA in LE	1.5 ± 0 ^b^	1 ± 0 ^c^	1.5 ± 0 ^b^	3 ± 0 ^a^	3 ± 0 ^a^
PCNA in GE	0 ± 0 ^c^	1 ± 0 ^c^	2 ± 0 ^b^	2.3 ± 0.6 ^b^	2.5 ± 0 ^a^
PCNA in S	0 ± 0 ^c^	0.5 ± 0 ^c^	1 ± 0 ^c^	1 ± 0 ^c^	3 ± 0 ^a^

Note: Means with different superscripts a, b, and c within the same row are significantly different from each other at *p* < 0.05 (n = 6 rats per group). G1 = Control, G2 = (IMC 4.33 mg/kg bwt), G3 = (IMC 4.33 + EBN 60 mg/kg bwt), G4 = (IMC 4.33 + EBN 90 mg/kg bwt), G5 = (4.33 + 120 mg/kg bwt). LE = Luminal epithelium, GE = Glandular epithelium, S = Stroma cell.

**Table 4 biology-14-00159-t004:** Mean score of E2R and P4R expression in uterine tissues of the rats exposed to IMC and augmented with EBN.

Parameters	Groups				
	G1	G2	G3	G4	G5
E2R in LE	0 ± 0 ^c^	1.5 ± 0 ^c^	1.6 ± 0.3 ^b^	3 ± 0 ^a^	3 ± 0 ^a^
E2R in GE	1 ± 0 ^c^	1 ± 0 ^c^	2 ± 0 ^a^	3 ± 0 ^a^	3 ± 0 ^a^
E2R in S	0.5 ± 0 ^c^	0.5 ± 0 ^c^	2 ± 0 ^a^	2 ± 0 ^a^	2.5 ± 0 ^a^
P4R in LE	0.3 ± 0.16 ^b^	0 ± 0 ^c^	0 ± 0 ^c^	2.5 ± 0 ^b^	3 ± 0 ^a^
P4R in GE	0 ± 0 ^c^	0 ± 0 ^c^	2 ± 0 ^a^	1 ± 0 ^c^	2.8 ± 0.16 ^a^
P4R in S	1.6 ± 0.16 ^b^	2 ± 0 ^a^	0.3 ± 0.3 ^c^	2.5 ± 0 ^a^	3 ± 0 ^a^

Note: Means with different superscripts a, b, and c within the same row are significantly different from each other at *p* ≤ 0.05 (n = 6 rats per group). G1 = Control, G2 = (IMC 4.33 mg/kg bwt), G3 = (IMC 4.33 + EBN 60 mg/kg bwt), G4 = (IMC 4.33 + EBN 90 mg/kg bwt), G5 = (4.33 + 120 mg/kg bwt). LE = Luminal epithelium, GE = Glandular epithelium, S = Stroma cell.

## Data Availability

The data presented in this study are available upon request from the corresponding author. The data are not publicly available due to privacy laws concerning the animals involved.

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
