# Peer review of "Edible Bird’s Nest (EBN) Ameliorates the Effects of Indomethacin (IMC)-Induced Embryo Implantation Dysfunction in Rats"

_biology, 2025, doi:10.3390/biology14020159_

Round 1
Reviewer 1 Report
Comments and Suggestions for Authors
This study focuses on the potential of IBN to reduce the toxic effect of indomethacin on the reproductive system of female rats.
However, there are some questions to the authors and comments.
1. The authors may have mistakenly referred to references 12 and 13 on the toxic effect of indomethacin on the reproductive system of females [line 88-91]. Could the authors indicate the works in which they investigated the influence of indomethacin on ovulation and implantation in rats?
2. Section 2.1 has reference 19, but there is no information on the determination of EBN doses. Could the authors provide a correct reference or describe the method? No method of administration of the IBN drug is specified.
3. Not specified how indomethacin administration was determined.
4. Rows 125-127, "the fertility index is shown in Table 1" but Table 1 does not contain this information.
5. How the authors explain the use of parametric statistical methods for such a small sample (6 observations in group)
6. In section 3.5, the authors show SEM images at 8 days of pregnancy, indicating pinopods as a sign of receptivity of the luminal epithelium of the rat uterus, however it is believed that implantation in rats occurs on the 5th day after fertilization. How do the authors explain this contradiction?
7. The thickness of the luminal epithelium is misrepresented in image 2b.
8. The authors do not show images of gross assessment of implantation of uterus for 8 days. Why?
9. At the 8th day of pregnancy, there should be signs of deciduation of the uterine wall that are not found in Figures 6-11. How the authors can explain this. Also, the authors do not indicate which parts of the pregnant uterus they have investigated.
10. In table 2, the authors indicated parameters of histopathological lesions. However, such signs may be characteristic of the uterine epithelium in the phases of estrus and metastus. Could the authors point to the microphoto the phase of an еstrous cycle or the presence of pregnancy?
Author Response
|
MDPI Biology paper corrections Reviewer 1
|
|||
|
This study focuses on the potential of IBN to reduce the toxic effect of indomethacin on the reproductive system of female rats. However, there are some questions to the authors and comments. |
|||
|
Comments |
Response |
Location |
|
|
1 |
The authors may have mistakenly referred to references 12 and 13 on the toxic effect of indomethacin on the reproductive system of females [line 88-91]. Could the authors indicate the works in which they investigated the influence of indomethacin on ovulation and implantation in rats? |
The references (both in-text and list) have been revised.
|
Line 71-77 |
|
2 |
Section 2.1 has reference 19, but there is no information on the determination of EBN doses.
|
In section 2.1: The doses were determined according to body weight in all rats.
Route of EBN administration has been mentioned |
Line 104-106 and 111-112
Line 106-107 |
|
3 |
Not specified how indomethacin administration was determined. |
Route of administration of indomethacin has been included in methodology |
Line 104-106 |
|
4 |
Rows 125-127, "the fertility index is shown in Table 1" but Table 1 does not contain this information |
It was a typing mistake the fertility index has been mentioned in section 2.5 |
Line 150-152 |
|
5 |
How the authors explain the use of parametric statistical methods for such a small sample (6 observations in group) |
We performed normality test for the data |
Line 219-223 |
|
6 |
In section 3.5, the authors show SEM images at 8 days of pregnancy, indicating pinopods as a sign of receptivity of the luminal epithelium of the rat uterus, however it is believed that implantation in rats occurs on the 5th day after fertilization. How do the authors explain this contradiction? |
In rats pinopods can be observed during pregnancy day 3-8 included a reference. Also published in our previous article by Albishtue et al. (2019).
|
Line 288-289
|
|
7 |
The thickness of the luminal epithelium is misrepresented in image 2b. |
It was a typing error I mentioned IMC enhanced LE thickness in a “dose-dependent manner instead of EBN |
Line 240-253 |
|
8 |
The authors do not show images of gross assessment of implantation of uterus for 8 days. Why? |
Images of gross assessment of implantation of uterus for day 8, have been included |
Line 257-264 |
|
9 |
At the 8th day of pregnancy, there should be signs of deciduation of the uterine wall that are not found in Figures 6-11. How the authors can explain this. Also, the authors do not indicate which parts of the pregnant uterus they have investigated. |
New images have been incorporated of uterine sample tissues of pregnant rats focusing on uterine layers (luminal and glandular epithelium and stroma and implantation sites) |
Line 297-377 |
|
10 |
In table 2, the authors indicated parameters of histopathological lesions. However, such signs may be characteristic of the uterine epithelium in the phases of estrus and metastus. Could the authors point to the microphoto the phase of an еstrous cycle or the presence of pregnancy? |
New images have been incorporated where the implantation site has been shown along with lesions indicated |
Line 269-282 |

Reviewer 2 Report
Comments and Suggestions for Authors
The reported aim of this study was to ascertain whether pre-treatment with the Edible Bird's Nest (EBN) will reduce IMC-induced toxicity in pregnant rats. The authors depended heavily on subjective qualitative morphological light microscopic assessments of uterine/endometrial histology which was complimented with IHC assessments of staining intestines of selected endometrial protein biomarkers (namely PR, ER, EGF, EGFR, VEGEF and PCNA) in histological sections of control and treated gravid uteri of Sprague-Dawley rats treated with increasing dosages of EBN (60, 90 and 120mg/kg, respectively) in the presence of a fixed dosage of Indomethacin (4.33mg/kg) for a period of 8 weeks.
The major weakness of this manuscript is the lack of quantitative assessments of the above-mentioned uterine/endometrial protein biomarkers in control and treated rats. This major oversight created a major deficiency in the present data sets that makes it hard to agree with most of the authors’ conclusions on a plausible reproductive benefit for the use of EBN to treat IMC-induced uterine/endometrial damage!
Major concerns:
1. As mentioned above, please provide quantitative assessments of the uterine/endometrial biomarkers you described in your present manuscript (namely PR, ER, EGF, EGFR, VEGEF and PCNA). For this purpose, the authors are encouraged to use Western blotting or ELISA kits to properly quantify the protein expression of these biomarkers. It is not sufficient to exclusively rely on subjective qualitative morphological light microscopic and IHC assessments of uterine sections as currently presented (e.g., lines 303- 382 and elsewhere in the manuscript).
2. According to the current materials and methods, the authors administered fixed amount of IMC (4.33mg/kg) to all experimental groups of rats! This experimental approach jeopardized the authors’ conclusions of a “dose-dependent” alterations in the uteri of treated rats (e.g., lines 402- 408 and elsewhere in the manuscript). The authors are strongly encouraged to reconsider their current conclusions in the face of the fact of their currently absent IMC dose-response studies.
3. Except for their use of the PCNA IHC staining, it is unclear how did the authors assess features of uterine/endometrial inflammation. There are currently no proteomic data or other specialized labeling of inflammatory reactions in this study to support the authors conclusions on this aspect (e.g., lines 175- 183 and lines 430- 445 and elsewhere in the manuscript).
4. Negative control staining for the IHC detections of PR, ER, EGF, EGFR and VEGEF in the uteri/endometria of all experimental rat groups are required. The current lack of these negatively stained slides compromises the credibility of the authors conclusions as presented in this manuscript.
5. Figure 2 and associated data: It’s unclear how was it possible for the authors to conclude that IMC enhanced LE thickness in a “dose-dependent manner” in treated rats (lines 251- 259 and bar-graphs in Figure 2) when they only reported the use of a fixed dose of 4.33mg/kg IMC in all treated rats?
6. Table 2 and associated data: it is unclear how did the authors assess vacuolation, atrophy, congestion and infiltration of inflammatory cells in the uteri of their experimental groups of rats? A detailed description of the methods used in obtaining these experimental parameters are needed.
7. There is currently no formal numerical assessment of the uterine expression of pinopodes in this study (e.g., lines 294- 302). The authors are strongly encouraged to perform and present formal statistical analysis of the expression of uterine pinpodes in their samples.
8. Vaginal smears: Lines 236- 242 and elsewhere in the manuscript: the ratios of the various cell types in vaginal smears should be calculated, expressed numerically as % of total per cell type and analyzed statistically. Graphic representations of these analyses are required in assessing reproductive responses of the vaginal epithelium in vaginal smears!
9. Lines 434-440; current citations # 43 does not reference the use of ICM but rather cadmium! Also, citation # 44 describes the solitary and isolated effects of ICM on the murine reproductive organs in the absence of EBN! Please retract and reconsider your present conclusions on this aspect (e.g., lines 434-440).
10. Lines 490-500: the authors never measured albumin levels in the serum/plasma of their experimental rats. It makes no sense arguing in favor of the potential influence of low serum albumin on the toxicity of IMC or the proposed mode of action of EBN in the absence of these experimental data!
11. Lines 185-193: the SEM sample preparation and image acquisition procedure need to be referenced appropriately.
12. Lines 134- 143 (EBN preparation): what is then final volume of the boiled EBN solution? Its presently unclear how much % of the raw 1gm/100ml of EBN was retained after the boiling process? Was an attempt made to concentrate the boiled emulsion to a standardized measurable quantity of EBN?
13. Figure 1: the bar graph depiction of body weight changes over time is very daunting as presented. Please consider reformatting it to be presented in tracible lines or whiskers instead.
Author Response
|
MDPI Biology paper corrections Reviewer 2 |
|||
|
Major concerns |
Responses |
Location |
|
|
1 |
As mentioned above, please provide quantitative assessments of the uterine/endometrial biomarkers you described in your present manuscript (namely PR, ER, EGF, EGFR, VEGEF and PCNA). For this purpose, the authors are encouraged to use Western blotting or ELISA kits to properly quantify the protein expression of these biomarkers. It is not sufficient to exclusively rely on subjective qualitative morphological light microscopic and IHC assessments of uterine sections as currently presented (e.g., lines 303- 382 and elsewhere in the manuscript). |
For this project we chose to conduct subjective qualitative morphological light microscopic and IHC assessments of uterine sections. However, we have used an appropriate scoring method for each variable to make the assessment more objective.
Nevertheless, we shall consider western blot and/or ELISA for protein expression of these biomarkers for our future experiments |
|
|
2 |
According to the current materials and methods, the authors administered fixed amount of IMC (4.33mg/kg) to all experimental groups of rats! This experimental approach jeopardized the authors’ conclusions of a “dose-dependent” alterations in the uteri of treated rats (e.g., lines 402- 408 and elsewhere in the manuscript). The authors are strongly encouraged to reconsider their current conclusions in the face of the fact of their currently absent IMC dose-response studies. |
The statement has been corrected as suggested |
Line 308-389 and Line 390-394 |
|
3 |
Except for their use of the PCNA IHC staining, it is unclear how did the authors assess features of uterine/endometrial inflammation. There are currently no proteomic data or other specialized labeling of inflammatory reactions in this study to support the authors conclusions on this aspect (e.g., lines 175- 183 and lines 430- 445 and elsewhere in the manuscript). |
Features of uterine/endometrial inflammation were assessed by using H&E technique using appropriate scoring methods for infiltration of inflammatory cells, vacuolation, congestion, atrophy |
Line 164-173 and
|
|
4 |
Negative control staining for the IHC detections of PR, ER, EGF, EGFR and VEGEF in the uteri/endometria of all experimental rat groups are required. The current lack of these negatively stained slides compromises the credibility of the authors conclusions as presented in this manuscript. |
NC template image have been included in all IHC results |
Line 322-372 |
|
5 |
Figure 2 and associated data: It’s unclear how was it possible for the authors to conclude that IMC enhanced LE thickness in a “dose-dependent manner” in treated rats (lines 251- 259 and bar-graphs in Figure 2) when they only reported the use of a fixed dose of 4.33mg/kg IMC in all treated rats? |
It was a typing error I mentioned IMC enhanced LE thickness in a “dose-dependent manner instead of EBN |
Line 240-248 |
|
6 |
Table 2 and associated data: it is unclear how did the authors assess vacuolation, atrophy, congestion and infiltration of inflammatory cells in the uteri of their experimental groups of rats? A detailed description of the methods used in obtaining these experimental parameters are needed. |
A detailed description of assessment method based on scoring has been given for the lesions found in each experimental groups of rats |
Line 164-173 |
|
7 |
There is currently no formal numerical assessment of the uterine expression of pinopodes in this study (e.g., lines 294- 302). The authors are strongly encouraged to perform and present formal statistical analysis of the expression of uterine pinpodes in their samples. |
I agree that quantifying the number of pinopods for each group by taking an appropriate number of samples would have provided valuable additional information. However, the SEM analysis was performed to observe the presence of pinopods to confirm endometrial receptivity during pregnancy day 4-8 Another reason was that the SEM analysis was costly |
Line 287-296 |
|
8 |
Vaginal smears: Lines 236- 242 and elsewhere in the manuscript: the ratios of the various cell types in vaginal smears should be calculated, expressed numerically as % of total per cell type and analyzed statistically. Graphic representations of these analyses are required in assessing reproductive responses of the vaginal epithelium in vaginal smears! |
This is part of the methodology we performed vaginal smears and the methods of calculating the cell count has been mentioned with reference |
Line 127-142 |
|
9 |
Lines 434-440; current citations # 43 does not reference the use of ICM but rather cadmium! Also, citation # 44 describes the solitary and isolated effects of ICM on the murine reproductive organs in the absence of EBN! Please retract and reconsider your present conclusions on this aspect (e.g., lines 434-440). |
The reference has been corrected and latest references cited accordingly
|
Line 420-422 |
|
10 |
Lines 490-500: the authors never measured albumin levels in the serum/plasma of their experimental rats. It makes no sense arguing in favor of the potential influence of low serum albumin on the toxicity of IMC or the proposed mode of action of EBN in the absence of these experimental data!
|
It was a mistake and has been removed from discussion part |
|
|
11 |
Lines 185-193: the SEM sample preparation and image acquisition procedure need to be referenced appropriately. |
References included to the SEM sample preparation and image acquisition procedure |
Line 174-183 |
|
12 |
Lines 134- 143 (EBN preparation): what is then final volume of the boiled EBN solution? Its presently unclear how much % of the raw 1gm/100ml of EBN was retained after the boiling process? Was an attempt made to concentrate the boiled emulsion to a standardized measurable quantity of EBN? |
The idea of heating EBN is to simulate the tradition of making EBN soup for human/animal consumption. We followed the previous publication (Albishtue et al., 2019) |
Line 121-123 |
|
13 |
Figure 1: the bar graph depiction of body weight changes over time is very daunting as presented. Please consider reformatting it to be presented in tracible lines or whiskers instead. |
A line graph has been incorporated |
Line 234-238 |

Reviewer 3 Report
Comments and Suggestions for Authors
The manuscript mainly designed and behaved the animal experiment to studied the effect of edible bird’s nest (EBN) in reducing IMC-induced toxicity in pregnant rats. And the authors found that EBN’s potential beneficial in anti-IMC-induced toxicity in clinical. The manuscript is recommended to be accepted by the journal Biology as the creativity and the workload of the manuscript are both enough. However, the writing of the manuscript can not reach the average level of the journal Biology. Here are some concerns for the authors:
1) In the Abstraction section, the authors directedly introduced the aim of the study. In fact, the authors may need to introduce the background and dig some scientific question around the background before introducing the work.
2) The Introduction section was written too chaos to understand. In fact, the authors are recommended to rewrite this section following the written of the “Simple Summary”.
3) Table 1 is not in need. The authors just need to introduce the 5 groups using sentences.
4) In the Materials and Methods section, the authors always loss to type a blank space between the number and the unit. This does not conform to the writing standards of English.
Comments on the Quality of English LanguageThe Introduction section should be rewrite.
Author Response
|
MDPI Biology paper corrections Reviewer 3
|
|||
|
Major concerns |
Responses |
Location |
|
|
1 |
In the Abstraction section, the authors directedly introduced the aim of the study. In fact, the authors may need to introduce the background and dig some scientific question around the background before introducing the work.
|
The abstract has been revised and a scientific question around the background has also been included |
Line 31-33
|
|
2 |
The Introduction section was written too chaos to understand. In fact, the authors are recommended to rewrite this section following the written of the “Simple Summary”. |
The Introduction section has been revised as suggested |
Line 51-97 |
|
3 |
In the Materials and Methods section, the authors always loss to type a blank space between the number and the unit. This does not conform to the writing standards of English. |
A blank space between the number and the unit has been considered |
Throughout the paper
|
|
4 |
Negative control staining for the IHC detections of PR, ER, EGF, EGFR and VEGEF in the uteri/endometria of all experimental rat groups are required. The current lack of these negatively stained slides compromises the credibility of the authors conclusions as presented in this manuscript. |
NC template images have been included in all IHC results |
All IHC figures 7 to 12 |

Round 2
Reviewer 2 Report
Comments and Suggestions for Authors
- Thank you for updating your graphs in Figures 1, 2 and 4, as well as providing NC slides. However, the major concern with this manuscript remains to be the lack of formal numerical assessment of the uterine expression of pinopodes in this study (e.g., lines 288- 296). The authors are once again strongly advised to perform and present formal statistical analysis of the expression of uterine pinopodes in their samples. Data obtained from this analysis are expected to clarify many physiological and functional aspects of the potential beneficial use of EBN to mitigate IMC-induced embryo implantation dysfunction in murine models. Please consider providing this kind of analysis.
- The newly added first sentence in the updated abstract needs to be re-phrased for easy English read.
- New Figure 4A: please provide SD bars for your bar-graphs or justify lack of thereof.
- Lines 288-289: the newly added sentence makes no sense! Please revise with more clarity.
Comments on the Quality of English LanguageThe English could be improved to more clearly express the research.
Author Response
|
MDPI Biology paper corrections Round 2 Reviewer 2
|
|||
|
This study focuses on the potential of IBN to reduce the toxic effect of indomethacin on the reproductive system of female rats. However, there are some questions to the authors and comments. |
|||
|
Comments |
Response |
Location |
|
|
1 |
Thank you for updating your graphs in Figures 1, 2 and 4, as well as providing NC slides. However, the major concern with this manuscript remains to be the lack of formal numerical assessment of the uterine expression of pinopodes in this study (e.g., lines 288- 296).The authors are once again strongly advised to perform and present formal statistical analysis of the expression of uterine pinopodes in their samples. Data obtained from this analysis are expected to clarify many physiological and functional aspects of the potential beneficial use of EBN to mitigate IMC-induced embryo implantation dysfunction in murine models. Please consider providing this kind of analysis. |
Thank you for your kind response. |
and Line 448-457 |
|
2 |
The newly added first sentence in the updated abstract needs to be re-phrased for easy English read. the influence of indomethacin on ovulation and implantation in rats? |
The first sentence in abstract has been rephrased |
Line 31-33 |
|
3 |
New Figure 4A: please provide SD bars for your bar-graphs or justify lack of thereof.
|
Figure. 4a. |
Line 268 |
|
4 |
Lines 288-289: the newly added sentence makes no sense! Please revise with more clarity.
|
The sentences related to SEM finding has been revised as suggested |
Line 290-292 |
Reviewer 3 Report
Comments and Suggestions for Authors
The manuscript has been revised according to the suggestions. The revised manuscript has been revised better than the 1st edition. The manuscript is recommended to be accepted by the journal Biology.
Author Response
|
MDPI Biology paper corrections Round 2 Reviewer 3
|
|||
|
This study focuses on the potential of IBN to reduce the toxic effect of indomethacin on the reproductive system of female rats. However, there are some questions to the authors and comments. |
|||
|
Comments |
Response |
Location |
|
|
1 |
The manuscript has been revised according to the suggestions. The revised manuscript has been revised better than the 1st edition. The manuscript is recommended to be accepted by the journal Biology. |
Thank you for your kind response and accepting our manuscript revisions for Journal Biology |
|
